# Analysis of the Stressed State of Sand-Soil Using Ultrasound

**Lukas Benedikt Schumacher** [1], **Mykola Sysyn** [1] , **Ulf Gerber** [1] and **Szabolcs Fischer** [2],*

[1]    Department of Planning and Design of Railway Infrastructure, Technical University Dresden, D-01069 Dresden, Germany
[2]    Central Campus Győr, Széchenyi István University, H-9026 Győr, Hungary
*    Correspondence: fischersz@sze.hu; Tel.: +36-(96)-613-544

**Abstract:** The maintenance of the ballast substructure is an important cost-driver for railway systems. The problem is that today's condition monitoring methods are insufficient to collect detailed data on the compaction and stress allocation inside the ballast bed. That makes it challenging to improve the maintenance technology and organization. This study aimed to investigate the applicability of the ultrasound method for analyzing the state of stress of sand-soil and the relation between the residual stress and wave propagation velocity. The experiments on the sand in a box with different allocations of the ultrasonic receivers and pressure measurement cells were produced under different external loading. In addition, the vertical and horizontal stress distributions were measured. The results showed a correlation between the test load, the state of stress, and the ultrasound propagation velocity. Moreover, the residual stresses after the loading cycles were analyzed.

**Keywords:** ballast bed; sand; non-destructive testing; stressed state; ultrasound propagation

## 1. Introduction

The railroad system is often discussed in society, politics, and expert groups since it is considered essential for future mobility concepts and climate protection. It increases the motivation to improve the systems' technical and economic efficiency.

This study focuses on the track substructure. It can be classified into two general designs: ballasted [1–4] and un-ballasted (ballastless) tracks [5–7]. Although the ballasted track is the conventional solution, it still has some advantages compared to modern un-ballasted systems. Ballasted tracks have a lower initial cost and lower noise emission, and the position and height of the track are easily corrected. Nevertheless, ballasted tracks have high maintenance costs, which is a significant disadvantage. Since the ballast bed is a flexible construction on which the track panel is floating, settlements can occur. The inhomogeneous share then causes deviations in the position and level of tracks, and if the corresponding limit values are reached, operational restrictions are necessary to ensure safety. The intensity of track settlement is affected by a variety of parameters, e.g., the appearing static and dynamic axle load, the type of sleeper that is used, inhomogeneous grain distribution, occurring excitation frequencies, moisture, ballast particle shape, the ballast grading curve, ballast material, and the pollution of the track bed [8]. Track geometry failures and deterioration of the ballast layer also have influence on the infrastructure operation costs due to traction energy loss [9].

The track geometry deterioration can be detected by using a measuring car or smaller trolleys [10]. Calculating the settlement and collecting information about the track bed geometry is possible. (However, it must be mentioned, the so-called DIC—Digital Image Correlation—method could be also available for this railway track geometry measurement procedure [11–14]). There are also some indicators leading to information about track bed pollution. Available technologies are, e.g., arrow height measurement, camera systems, and ground penetration radar. The track geometry correction is realized by regular tamping

operations about every 4–5 years and less frequently, by cleaning or complete renewal of the track [8,15].

It can be stated that ballast maintenance, especially the often-performed tamping process, is a cost driver and factor for the economic efficiency of railway systems. Thus, the research of the factors influencing the deterioration process has an optimization potential for maintenance improvement [16]. The following paragraphs provide a literature review of the state of the science.

The detection of deterioration can be improved by using modern, non-destructive, and automatable measuring methods. In [17], different non-destructive options were compared. The high potential was ascribed to ground penetrating radar, falling weight deflectometer, and the impulse response principle.

The maintenance process itself is the object of investigation by many publications. DEM (discrete element method) is a standard method to investigate optimization potential: the study [18] showed that the efficiency of tamping varies for different sleeper types and tamping pick positions. The study [19] showed that the different pick-tamping phases have optimal frequencies and penetrations speeds, and [20] confirmed, among other findings, that the most effective vibration frequency for ballast is 35 Hz. Application of numeric simulation methods like DEM and FEM [2–4,21,22] could potentially improve the understanding of the mechanical processes in railway tracks. In [21], experiments showed that the pollution degree impacts the pick-tamping's effectiveness. In [22], different tamping methods were compared, and the effectiveness of side tamping was analyzed in an experimental setup using FEM (finite element method) and photogrammetry.

The present study focuses on detecting deterioration since this is mandatory to collect more data on the track condition, compare the efficiency of maintenance technologies, and improve the maintenance cycle. It was decided to investigate the ultrasound method in an experimental setup since it is one of the most common non-destructive methods in other disciplines like medicine or materials science. Furthermore, in [23] a disk transducer for measuring elastic waves on coarse-grained material as a geotechnical application was developed.

The relation of stress in granular media to elastic wave propagation is considered in the following papers. The influence of the stress history on wave velocities is studied in [24,25] using DEM simulations. The results show that P- and S-wave velocities increase under oedometric compression with confining pressure following a power law; the wave velocities vary slightly with the input frequency.

The papers [26,27] present experimental studies of elastic wave velocities in soil samples with different void ratios and stresses. The results showed that the increment of the normal stress component significantly influences compression wave velocity compared to shear wave velocity. The relation to the confinement stress is linear.

The studies [28,29] present numeric investigations of factors influencing wave propagation in dry granular materials with the help of DEM modeling. The elastic moduli and Poisson's ratio of each packing were obtained by compression using pressure and shear wave velocities. A linear relationship has been identified between the coordination number normalized by contact force and the elastic moduli normalized by confining pressure. Furthermore, it was obtained that an increase in the aspect ratio of particles leads to a notable increase in the elastic shear and pressure wave velocity. In contrast, for non-spherical particles with a given aspect ratio, an increased particle blockage causes a moderate reduction in wave velocity.

DEM modeling to simulate triaxial compression experiments was used in the study [30]. The spherical particles with four samples isotopically confined were applied at various initial packing densities and then sheared monotonically up to the critical state. The results showed that the major principal stress influences pressure wave, whereas the geometric mean stress and the mean coordination number influence the shear wave velocity more.

A shear-wave velocity-based constitutive model with critical state soil mechanics is presented in [31] to predict the undrained triaxial behavior of fine-grained sediments.

The laboratory tests were done for sediment samples ranging from silt-predominant to clay-predominant sediments. A power function was supposed to describe the relationship between mean effective stress and shear-wave velocity. Most of the presented studies are based on DEM modeling, which is characterized by an intensive calculation process that limits its application to relatively small samples.

Other approaches to numerical wave propagation studying the whole superstructure are described in [32–34]. The presented studies on wave propagation relation in granular media showed the homogenous stress distribution. The railway ballast is subjected to the locally inhomogeneous stresses that cause the corresponding wave propagation effects.

Application of elastic under sleeper pads (USP) or unde-ballast mats UBM mats [35–39] could potentially have a high impact on stress distribution and, first of all, on accumulation of the residual stresses in the ballast layer. The study [40] presents an estimation and explanation of the mechanism of the residual stress accumulation in the ballast layer after the cyclic loading.

The presented literature review demonstrates some approaches for ultrasound testing of granular soils. However, the approaches are different measurement and excitation systems that on one side, cannot be exactly replicated in further studies and, on the other side, are not suitable for the present engineering application. Thus, the development of the ultrasonic measurement system for railway ballast testing is necessary. Another shortcoming of the previous studies on wave propagation's relation to soil stresses is that the studies usually consider the homogenous stress distribution. However, the railway ballast is subjected to locally inhomogeneous stresses.

The present research aims to study the relation of the wave propagation velocity in the sand to the stressed state distribution in the medium using the developed ultrasonic measurement system and stress measurement one. In addition, the aspects of the local inhomogeneity of the wave propagation and stress distribution are considered. The sand material was used to test the system and the fundamental relations before the future application to the real ballast material.

## 2. Laboratory Measurements

The measurement setup consisted of three systems: the externally controlled loading device, the ultrasound measurement system, and the stress measurement system.

As a pressure hull, a wooden box (inside dimensions: $35 \times 35 \times 40$ cm) with a stamp was constructed and mounted in a ZWICK HB 160 servo-hydraulic press, in which the external loading cycle was performed. The box was filled with sand and contained the ultrasonic transmitter, ultrasonic receivers, and pressure measurement cells. To allow the transmitter to work independently of the test load, it was installed in a cavity in the stamp supported by a spring. The compression way and the adjusted test load were recorded.

The sand-soil material consists of angular quartz particles with size 0.5–1.5 mm with the bulk density 1650 $kg/m^3$ and the friction angle $32°$. The material was considered in the dry state.

It was decided to use a NBL45402H-A transducer (h: 53.5 mm, Ø45 mm) with a maximum allowable power of 50 W and a weight of 240 g because it is a robust system that allows for generating high impulse amplitudes. Usually, transducers of this type are used as cleaning equipment. Ultrasonic receivers of type A-14P20 (h: 6 mm, Ø14 mm) were located in five positions inside the box: in the sidewalls and on the bottom. For this measurement, only three of the five sensors were used. The sensor type was selected for the measurement due to its small size, low purchase price, and adequate performance in prior experiments.

The electronic transmitting unit was used to create ultrasonic impulses and consisted of an Arduino Uno microcontroller and an amplification system to power the transmitter with impulse voltage 0 . . . 500 V. The impulses had a 13 μs length and were emitted every 50 ms. The signal was transported to the transmitter using a shielded cable. The ultrasound sensors received the ultrasound signal, transferred to the receiving unit by a shielded cable,

amplified, and then recorded with an L-CARD E-502 DAC-system connected to a computer. The sampling rate was set to 500 kHz and the sampling time to 0.6 s. Two channels were used simultaneously: one was connected to the transmitting unit to record the transmitting signal, and the other was used to sample one ultrasound sensor. Although it would have technically been possible to record multiple signals in one measurement, it was decided to sample them sequentially to reduce problems with electromagnetic interferences. Figure 1 shows a schema and a drawing of the components of the measuring system. Figure 2 shows photos of the transmitting and receiving unit.

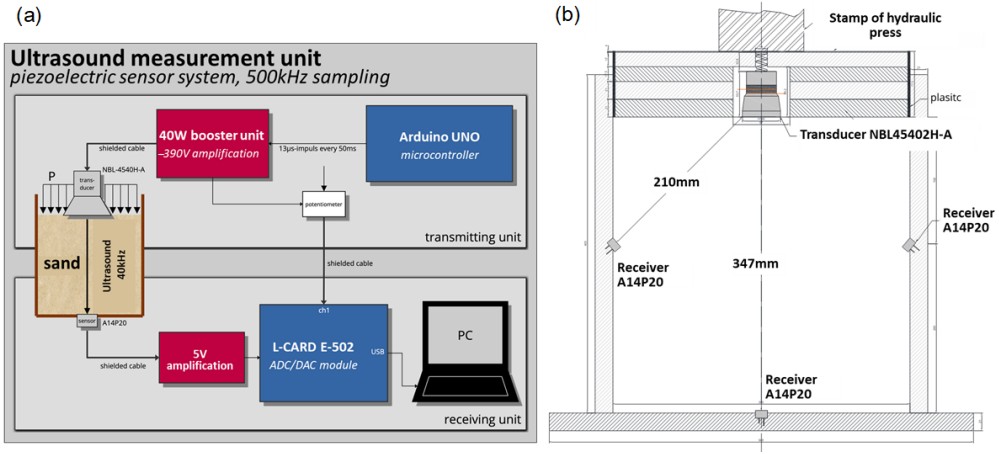

**Figure 1.** (**a**) Schema of the ultrasound measuring system, (**b**) Drawing of the test box with ultrasonic transmitter and receivers.

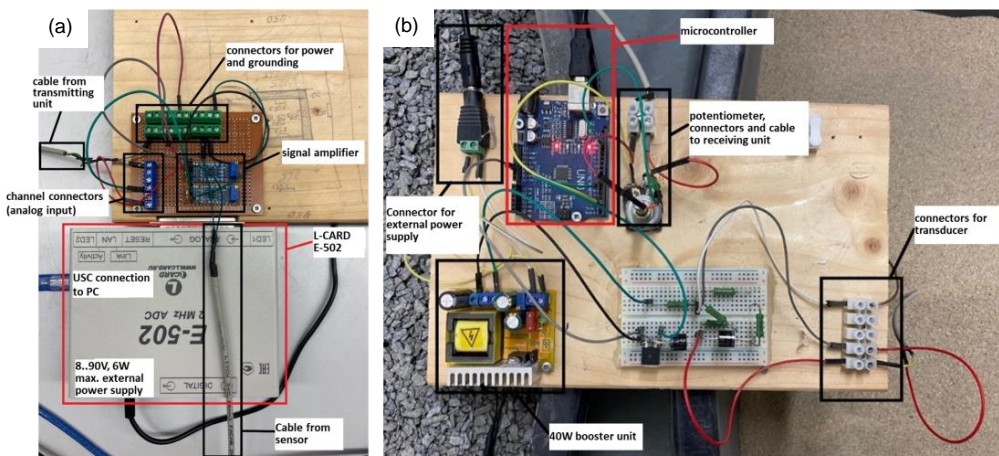

**Figure 2.** (**a**) Receiving unit with LCARD E-502 system, (**b**) Transmitting unit with microcontroller and booster unit.

The stress measurement system was realized by eleven pressure measurement cells based on strain gauge technology installed inside the box. A DAQ system QuantumX was used to calibrate and read the pressure cells. Figure 3 shows the stress measurement system. In the drawing on the right side, it becomes apparent that the pressure has been measured at six positions, in most cases in both the horizontal and the vertical direction.

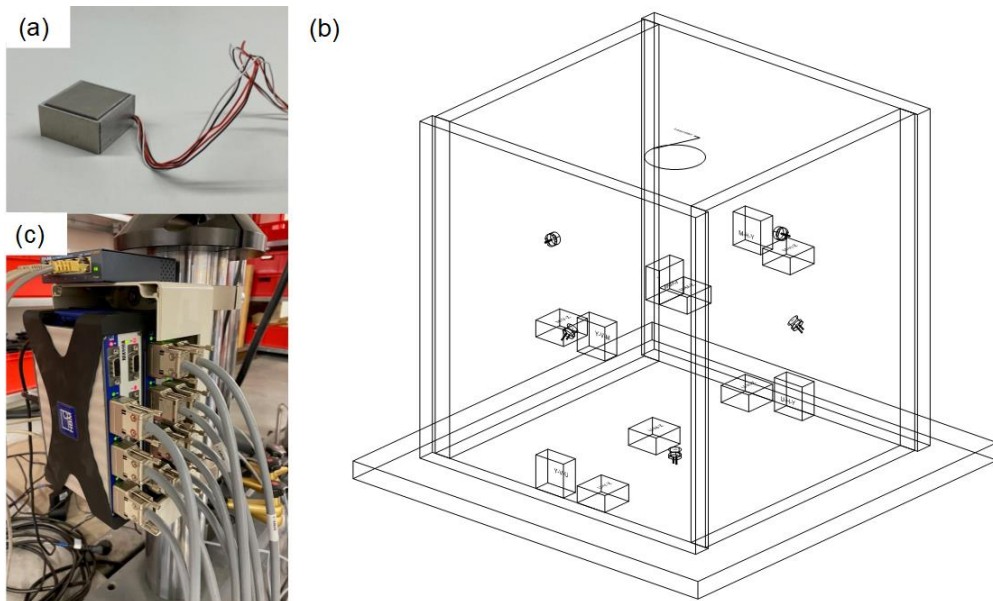

**Figure 3.** (**a**) Pressure measurement cell, (**b**) 3D model of the test box with sensor locations, (**c**) QuantumX DAQ system.

Figure 4 shows the experimental setup during the test procedure. On the left side the servo-hydraulic press with its stamp and under it the box is pictured as well as the operator station with the computer and the electronic equipment. On the right side, the inside of the box is shown with the pressure and the ultrasound sensors. The central pair of pressure sensors were held in place using a mesh wire.

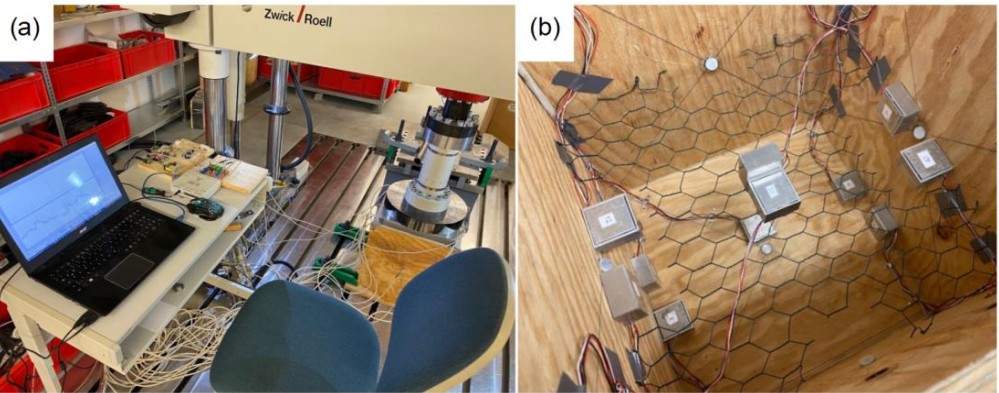

**Figure 4.** (**a**) Measuring station at the ZWICK HB 160 testing machine, (**b**) Interior view of the test box.

The test program started with a consolidation phase in which the sand was shacked at frequencies between one and 20 Hz. After that, the two main test cycles were performed with variations of the test load between zero and 2.5 kN, as shown in Figure 5.

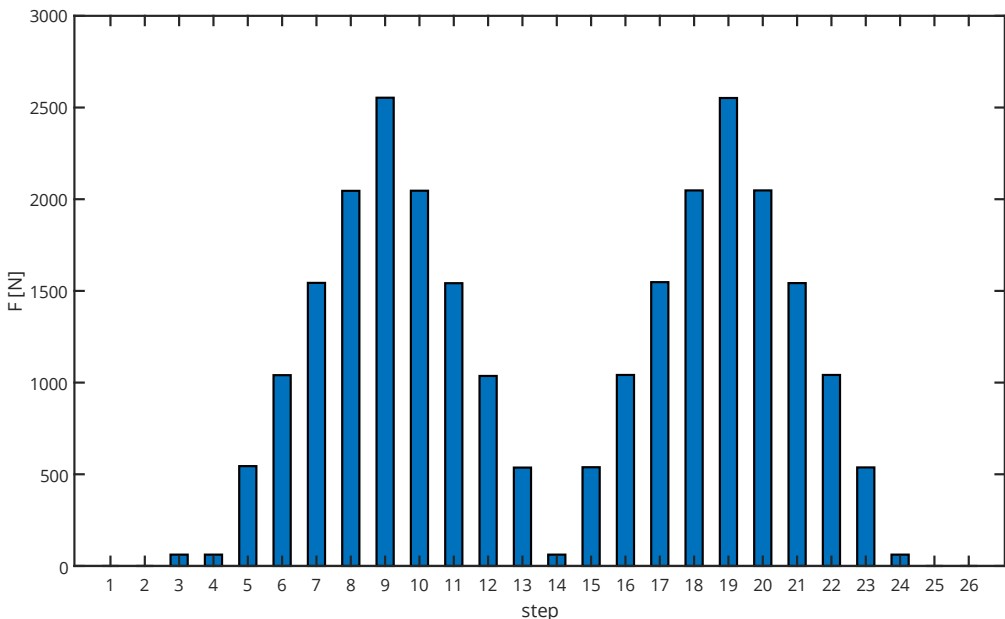

**Figure 5.** External loading steps for the sand test cycles with loadings up to 2500 N.

The recorded ultrasound data were preprocessed and analyzed using MATLAB [41]. By smoothing the signal, calculating its first derivative, and identifying the instant of time of the impulse and its receiving signal, it was possible to calculate the wave propagation time. Figure 6 shows a typical recording of piezo receivers under two different loadings. First, the starting impulse leaves a short electromagnetic impact visible on the chart's left side. Then, the signal is received after a break corresponding to the propagation time.

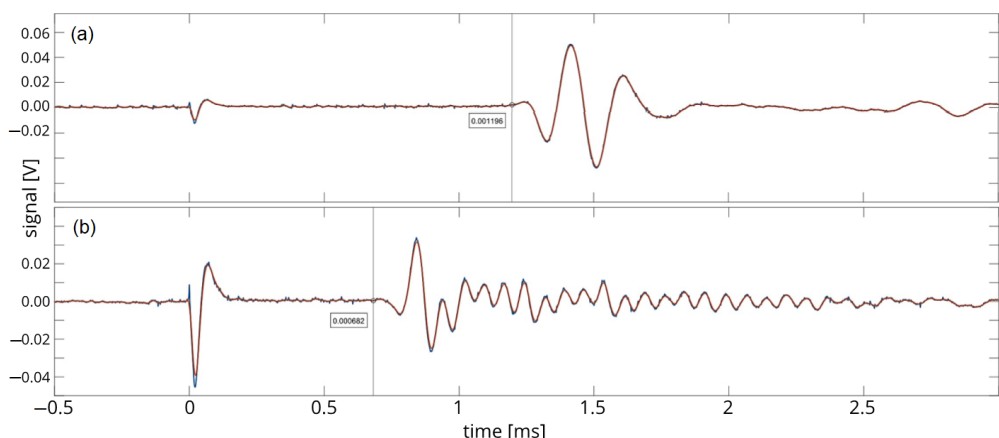

**Figure 6.** Examples for ultrasound measurement records with (**a**) 2000 N and (**b**) 2500 N loading.

## 3. Analysis and Interpretation

### 3.1. Local Pressure

Figure 7 compares the vertical and horizontal tensions that were measured for each step of the test program using diagrams. All the vertical tensions showed a substantial increase in response to an amplification of the test load, while the horizontal tensions on the bottom remained almost constant. On the first level ($h_1$ = 0.2 m) the horizontal tensions acted as inversely proportional to the test load. This effect can be highlighted by calculating the mean vertical and horizontal tensions and plotting them over the test load, like in Figure 8. Figure 9 visualizes the tension distribution in 3D, which helps understand local dynamic variations of tension. On the bottom, the highest tension was measured in the vertical direction in the center. At this position, the highest values without loading

appeared, but the increase due to the load was the smallest among the vertical tensions. At the front and the rear position on the bottom, the vertical and the horizontal tensions remained low. On the first level, the opposite situation arose. The highest tensions were measured on the rear and the front.

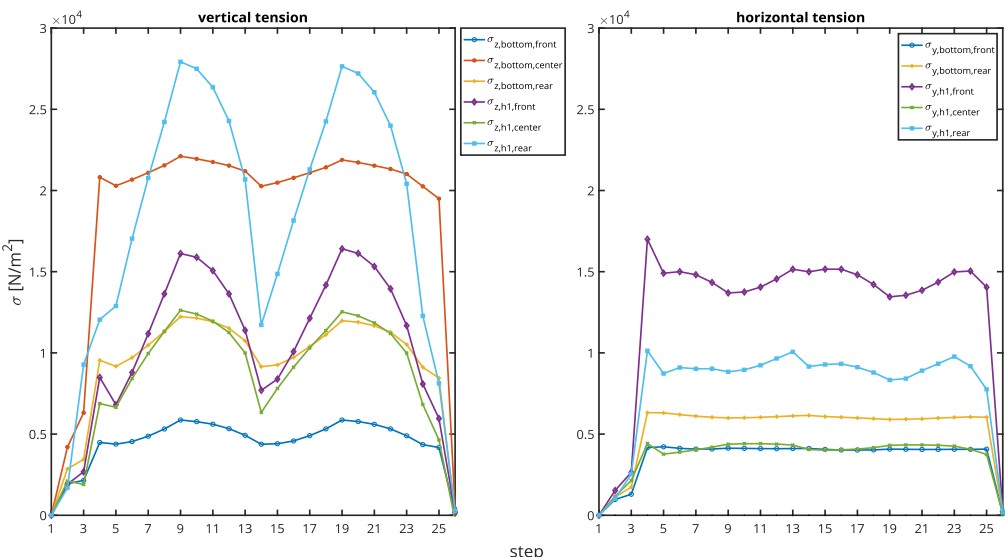

**Figure 7.** Vertical and horizontal pressure values in the box for each step of the test program.

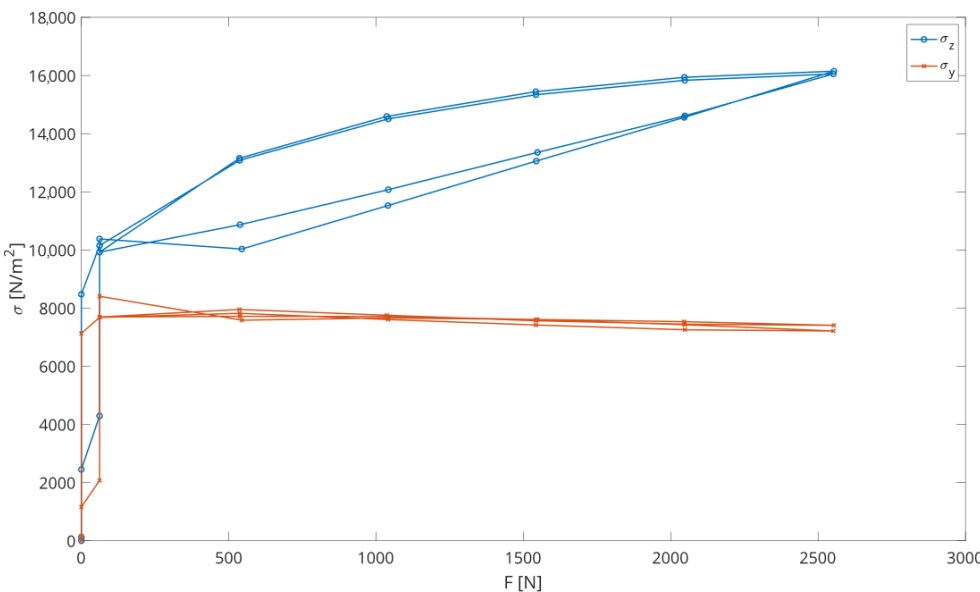

**Figure 8.** Mean vertical and horizontal pressure depending on the external loading.

The 2D representation of the relative stresses in sand during the first load cycle for both vertical and horizontal stresses in the cross-section of the box is presented in Figure 10. The color intensity of lines corresponds to the stress intensity (with the darkest shade corresponding to the 2500 N level). Figure 10a shows that the vertical stresses increase together with the increment of the external loading, and stay relatively homogenous in the middle measurement plane. However, the inhomogeneity of three local vertical stress distribution appear at the bottom measurement plane. Thereby, the horizontal stresses distribution (Figure 10b) on the middle plane has a similar inhomogeneity as the vertical stresses, but almost no relation to the vertical loading.

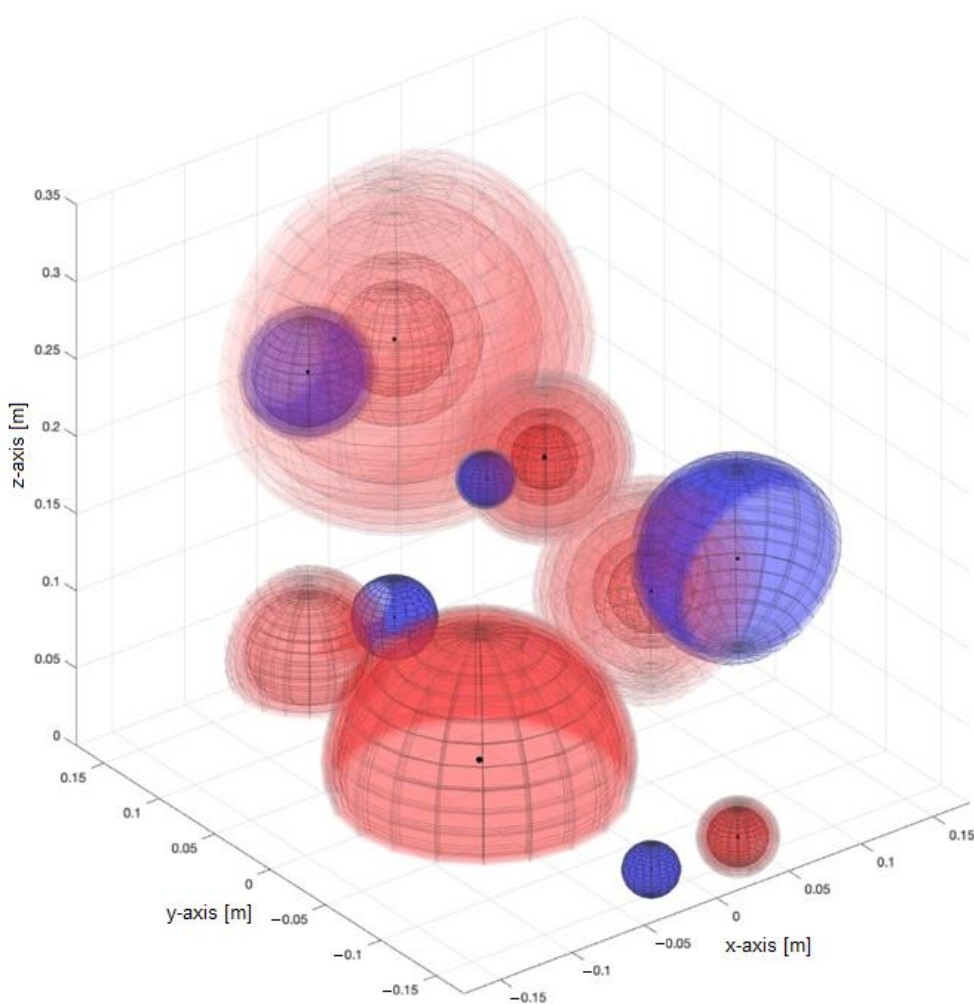

**Figure 9.** 3D presentation of the pressure in the sand for all loading levels (red: vertical, blue: horizontal, black dots: center points of the spheres).

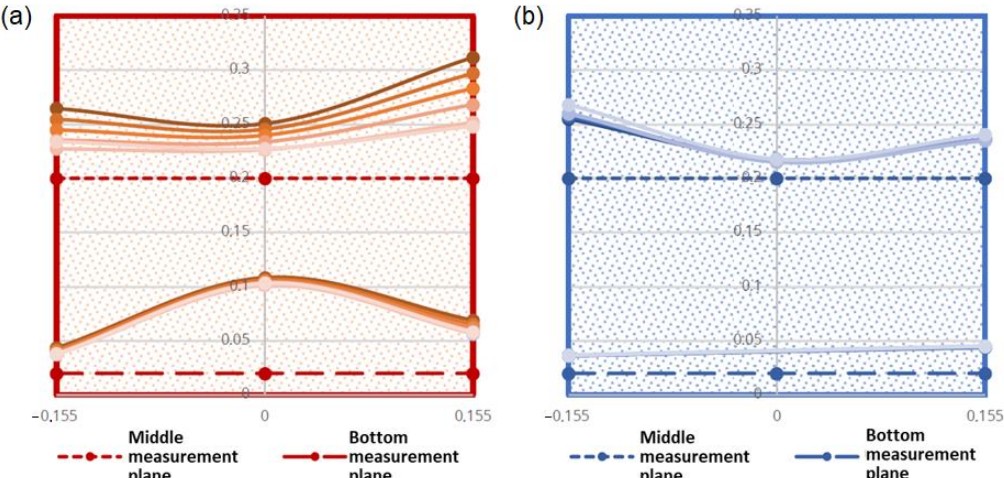

**Figure 10.** Two dimensional representation of the stresses in sand during the first load cycle: (**a**) vertical stresses, (**b**) horizontal stresses.

Figure 11 shows the path measurement from the servo-hydraulic press as a function of the test load. The figure contains all loading and unloading steps. Since no hysteresis or trends are visible, it indicates a linear relation between test load and compaction.

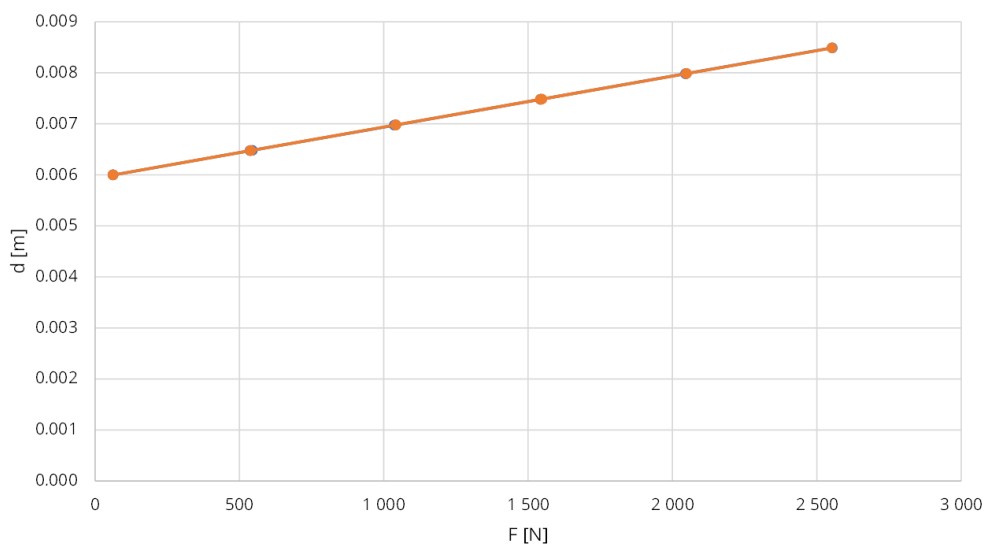

**Figure 11.** Measurement of the compaction path by the servo-hydraulic testing machine.

Some more information can be gained by analyzing the residual stress distribution. Since pressure has been measured with zero loading at different steps of the test cycle, these steps can be compared. Of interest is how much the residual tensions increased due to the shaking ($\sigma_4$-$\sigma_3$), after the first cycle ($\sigma_{14}$-$\sigma_4$) and after the second cycle ($\sigma_{24}$-$\sigma_{14}$). The results are displayed in Figure 12. Again, high values were calculated for the bottom plate's center and front and rear walls on the first level. It is interesting to note that the remaining horizontal tensions were dominant on the sidewalls. It can be assumed that a settlement appeared in the center, which would explain the high residual tension in this spot and that it reacted less dynamically compared to the other vertical tensions. The remaining high tension on the first level's sidewalls supports the assumption that the sidewalls absorbed much mechanical energy through friction.

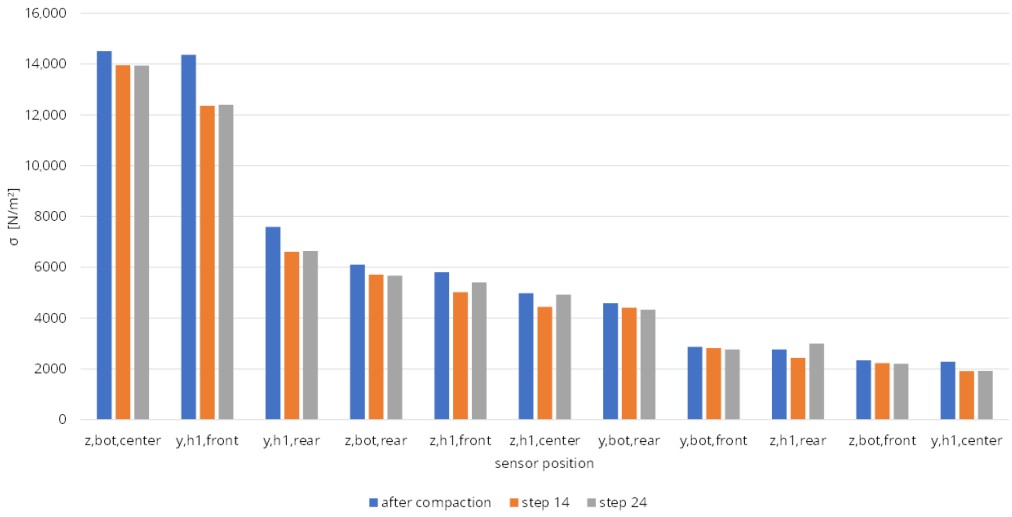

**Figure 12.** Distribution of residual pressure in the sand for the zero load steps of the test program.

These observations lead to some conclusions regarding mechanical processes inside the box:

- The energy from the stamp was converted into a vertical, elastically movement of sand soil, as shown in Figure 11.
- On the sidewalls, the movement energy was mainly consumed by friction with the wooden walls since high vertical dynamic in combination with pronounced horizontal tensions appeared in these spots. In the center, the movement energy caused a high material pressure which can be concluded from the reduction of the dynamic and the rising static share in tension from top to bottom.
- The distribution of tension was inhomogeneous.

### 3.2. P-Wave Propagation Velocity

As shown in Figure 6, the transit time between the ultrasound impulse and the received signal was measurable. The distance between the transmitter and sensor was calculated for each step of the test program considering the known geometry of the box and the current measured compression (Figure 11). From these two quantities, it was possible to calculate the wave propagation velocity c and display it over the measured test load in Figure 13. A correlation c(F) between the test load and the wave propagation velocity becomes visible. The correlation coefficient between F and c depends on which receiver and which phase of the test program is considered. The average value is 0.9478; the lowest is 0.8565, and the highest is 0.9826. A hysteresis between loading and unloading is on hand. The hysteresis can also be observed in Figure 8, which shows the mean vertical and horizontal tensions in dependence of the test load. It indicates that a connection between the state of stress and the p-wave velocity is likely to exist. However, it was not possible to find a quantifiable correlation c(σ) between the state of stress of sand and the p-wave speed, because there were not enough sensor positions to understand the complex stress allocation inside the box.

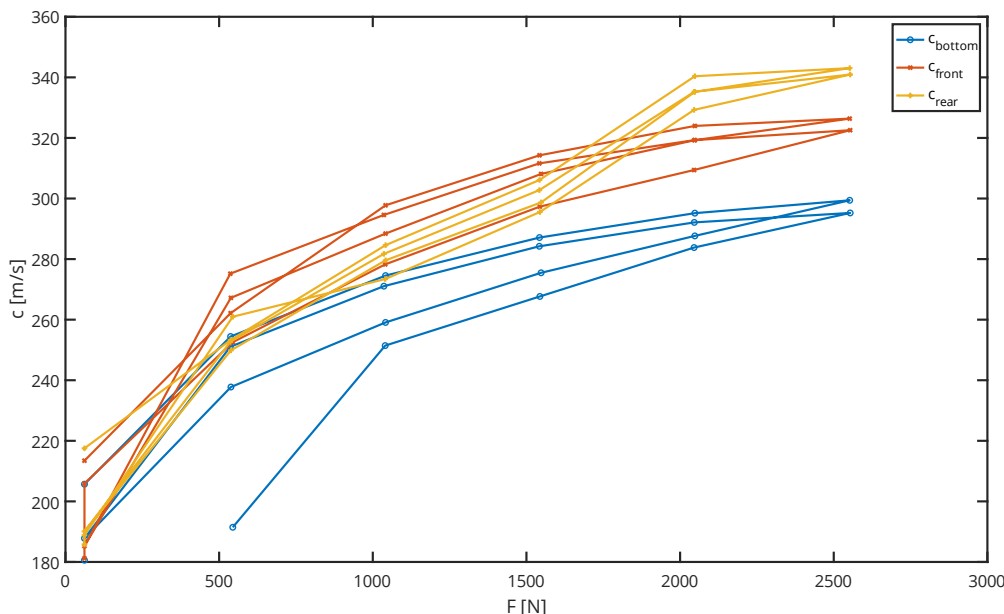

**Figure 13.** P-wave velocity depending on the test load.

### 4. Discussion

Developing a non-destructive method for testing ballast compaction quality and thus reducing the deterioration process using optimizing the maintenance technologies has great potential for saving the maintenance costs of railroads.

The experiments showed a strong correlation between the applied load on the test material and the ultrasound wave propagation speed inside the material, which corre-

sponds to the studies [24,26–31]. However, unlike previous studies, the determined relation of the vertical wave propagation to the stresses and loadings is not linear (Figure 13) in the initial loading region. The relatively low external loading can explain the behavior compared to the gravitational one that initially causes the growing vertical stress and wave velocity distribution. After the external loading increases, the differences in vertical stress distribution decrease. It causes a more linear relation to the vertical wave propagation velocity. Another reason for the nonlinear behavior could be the inhomogeneous stress distribution in the horizontal direction. The distributed stress measurements have shown a high inhomogeneity of stresses at the bottom of the box despite the relatively homogenous pressure application at the top surface of the sand layer. An additional research finding concerns the loading cycles: residual stresses accumulate in the sand sample during the loading cycles and wave propagation velocity increased in the unloaded state. The highest stress accumulation was detected in the central part of the box bottom. The sand friction interaction can explain the local stress inhomogeneity and the residual stress accumulation to the vertical walls of the box, as well as the elastic reaction of the walls. Similar behavior of the residual stress accumulation is noted in the other studies on ballast interlocking [40].

Considering this finding, it can be assumed that there is also a correlation between the stress of the material and the measured wave propagation speed, which can be detected by optimizing the experimental setup. This correlation will be the research objective of further experiments with real railway ballast material. It could then be used to obtain detailed information about the inner condition of ballast track beds without retrieving samples. It is the requirement for developing a non-destructive method for ballast testing. A long-term objective of these laboratory studies is to develop a method that can be used directly on railroads and is in step with actual practices. Therefore, a tomography of the railroad substructure is a medium-term goal based on the current methodical studies.

The research is intended to develop the methods for the coarse-grained railway ballast layer in future studies. In further experiments, it is intended to obtain the σ(c) curve for different materials by changing the experimental setup. Different approaches for technical improvements, such as changing the box's geometry and the pressure hull's material, using anti-friction coating, and collecting more data are applicable. The observations on the residual tensions indicated that high friction values appeared. A more homogenous stress distribution will be achieved if the friction is reduced. Additionally, a DEM-Simulation could lead to further insights.

## 5. Conclusions

The following conclusions can be stated based on the results of the present paper:

- Internal pressure in the land layer influences the pressure wave propagation velocity. Increase of the pressure from 2 to 23 kPa at the bottom of the box results in an increase in the vertical wave velocity from 180 to 360 m/s. The relation between ballast pressure and wave propagation velocity is nonlinear.
- The vertical stress distribution over the ballast box is subjected to high local inhomogeneity with up to two times the stress concentration in the central part of the box bottom.
- The residual pressure appears at the bottom of the ballast box and accumulates after the loading cycles. The residual stresses amount to up to 60% of the maximal ones.
- The residual pressure has an influence on the wave propagation velocity.

**Author Contributions:** Conceptualization, L.B.S., M.S., U.G. and S.F.; methodology, M.S. and U.G.; software, L.B.S. and M.S.; validation, L.B.S., M.S., U.G. and S.F.; formal analysis, L.B.S. and M.S.; investigation, M.S. and S.F.; resources, M.S.; data curation, L.B.S. and M.S.; writing—original draft preparation, L.B.S., M.S., U.G. and S.F.; writing—review and editing, L.B.S., M.S., U.G. and S.F.; visualization, L.B.S. and M.S.; supervision, M.S., U.G. and S.F.; project administration, M.S. and U.G.; funding acquisition, L.B.S., M.S., U.G. and S.F. All authors have read and agreed to the published version of the manuscript.

**Funding:** This research received no external funding.

**Data Availability Statement:** Not applicable.

**Acknowledgments:** The authors acknowledge the LCard Company for kind supports with measurement instruments.

**Conflicts of Interest:** The authors declare no conflict of interest.

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
