# Peer review of "Analysis of the Stressed State of Sand-Soil Using Ultrasound"

_infrastructures, doi:10.3390/infrastructures8010004_

Round 1

Reviewer 1 Report

1、 Physical properties of sand sample, including the material, the particle size, the moisture content and so on, have considerable impacts on the test results, they should be introduced in more detail.

2、 Due to ballast aggregates has a wide grading range (2.5mm <partical size<70 mm), whose mechanical behavior is significantly different from the sands, the conclusions in this manuscript cannot be extended to for railway ballast. In my opinion, this research is helpful to subgrade state detection using ultrasound, but has a little potential value for railway ballast.

3、 I strongly recommend that the authors reedit this manuscript aiming at the sand-soil material, NOT the coarse-grained ballast granular ballast bed.

Author Response

See attached pdf file.

Reviewer 2 Report

I suggest improving the graphical quality of the drawings and their descriptions (e.g. Fig. 1/2).

The article presents the parameters of the ultrasound measuring system in detail (e.g. Tab. 1). Similar parameters should be provided for the structure of the analyzed track.

1) What are the parameters of the ballast used in the tests presented in the article (according to EN 13450)?

2) Were USP pads or UBM mats used in the construction of the track?

If, for example, USP pads or UBM mats were not present in the analyzed track structure, it should be written that these elements are not analyzed in the article, but may have an impact on the stress in ballast, as presented in the works - for example:

1) https://doi.org/10.1016/j.trgeo.2019.01.005

2) https://doi.org/10.1007/s10035-018-0795-0

3) https://www.mdpi.com/1996-1944/14/2/313

4) A. de O. Lima, M. S. Dersch, Y. Qian, E. Tutumluer & J.R. Edwards Laboratory mechanical fatigue performance of under-ballast mats subjected to North American loading conditions

5) Indraratna, B., Nimbalkar, S., Navaratnarajah, S. K., Rujikiatkamjorn, C. & Neville, T. (2014). Use of shock mats for mitigating degradation of railroad ballast. Sri Lankan Geotechnical Journal - Special Issue on Ground Improvement , 6 (1), 32-41.

I believe that the work deserves publication in Infrastructures, after the Authors have taken into account the suggested comments and modifications in the article.

Author Response

See attached pdf file.

Reviewer 3 Report

This paper presents a novel laboratory test to characterize the mechanical behaviors of sands compaction. The results showed a correlation between the test load, the state of stress, and the ultrasound propagation velocity. Moreover, the residual stresses after the loading cycles were analyzed.

There are also some detailed comments for the authors.

1. Whether the sensor volume and stiffness will affect the test results?

2. Will the mesh wire covering the bottom and the surrounding wall surface affect the sand displacement, similar to the effect of the geogrid?

3. Since the author mentioned the local inhomogeneous of the sample, how can the results of this test explain its representativeness?

4. Please provide more interesting results or analysis, thanks.

Author Response

See attached pdf file.

Round 2

Reviewer 1 Report

The manuscript had been revised by authors considering reviwer's comments, and the preconditions of this study are declared properly. 

Reviewer 3 Report

Can be accepted